# Bayesian Inverse Problems Meet Flow Matching: Efficient and Flexible Inference via Transformers

## Abstract

In this paper, we present a new framework that combines Conditional Flow Matching (CFM) with a transformer-based architecture. This enables us to sample fast and flexibly from complex posterior distributions when solving Bayesian inverse problems. The methodology directly learns conditional probability trajectories from the data, leveraging CFM's ability to bypass iterative simulation and transformers' capacity to process an arbitrary number of observations. The primary outcomes show that relative parameter recovery errors are as low as 1.5%, and that inference time is reduced by up to 2,000 times on a CPU compared to the Markov Chain Monte Carlo, as demonstrated by three Bayesian problems.

## 1 INTRODUCTION

Numerous natural and engineering systems are amenable to rigorous mathematical modelling; however, the governing parameters of these models commonly remain latent and must be inferred from limited, noisy measurements. Bayesian inversion frames this problem as the estimation of the posterior distribution of the parameters conditioned on the observed data and a prior distribution, thereby delivering both point estimates and the attendant uncertainty quantification [Cotter et al., 2009, Koval et al., 2024]. Classical sampling schemes—most notably Markov chain Monte Carlo (MCMC) methods [Geyer, 1992]—are asymptotically exact, yet they typically require thousands of forward model evaluations per data set, rendering them impracticable for real-time digital-twin deployments [Kapteyn et al., 2021]. Consequently, recent research has pivoted toward amortised generative surrogates: variational auto-encoders (VAEs) [Kingma and Welling, 2022], generative adversarial networks (GANs) [Goodfellow et al., 2014], and diffusion

models [Sohl-Dickstein et al., 2015] are capable of representing intricate posterior landscapes, albeit only approximately, because they lack exact likelihood evaluation. In contrast, continuous normalizing flows preserve exact likelihood computation while dispensing with the costly iterative sampling loop [Gudovskiy et al., 2024], thereby constituting a promising foundation for scalable Bayesian inverse solvers.

### Our contribution

- We formulate the Bayesian inverse problem as the problem of learning the conditional probability distribution from samples, that can be easily constructed.

- We propose a transformer-based Conditional Flow Matching (CFM) Lipman et al. [2023] architecture that can handle different number of observations.

- We test our method on several inverse problems and compare it to the MCMC approach.

## 2 BACKGROUND AND RELATED WORK

**Classical Bayesian Inference**  Sampling–based schemes such as MCMC, Hamiltonian/Sequential Monte Carlo, and variational methods remain the work-horse for Bayesian inverse problems but become prohibitive in high dimensions. Each posterior sample often requires many expensive forward solves and derivative evaluations, which hampers real-time inversion and continual data assimilation.

**Deep-Learning Surrogates**  Physics-informed invertible neural networks (PI-INNs) [Raissi et al., 2019, Guan et al., 2023] replace the forward model with an explicit bijection, giving exact likelihoods yet demanding a hand-crafted loss for every PDE and a fixed sensor layout. Diffusion–transformer hybrids such as Simformer [Cunningham et al., 2024] scale to complex priors but rely on stochastic sampling, limiting interpretability and latency. In medical

imaging, score/diffusion generators have shown strong reconstruction quality [Aali et al., 2023, Song et al., 2022] but still lack closed-form posteriors.

**GANs and VAEs** MCGAN [Mücke et al., 2022] accelerates MCMC by learning a GAN surrogate, and MDGM [Xia and Zabaras, 2022] embeds a multiscale VAE prior inside MCMC. These hybrid designs improve prior expressiveness yet inherit the iterative sampling burden and assume a fixed number of observations.

**Continuous Normalizing Flows** Flow Matching (FM) [Lipman et al., 2023] trains continuous normalizing flows via vector-field regression, unifying diffusion and optimal-transport paths for rapid likelihood evaluation and ODE-based sampling. When combined with variational inference [Whang et al., 2021], CNFs yield flexible posteriors but incur high memory-time costs as system size grows.

Our Conditional Flow-Matching Transformer (CFM-Tr) inherits the exact likelihood estimation and training stability of FM while using attention to ingest an arbitrary, unordered set of measurements. This removes the fixed-sensor limitation of PI-INNs, avoids iterative latent sampling required by GAN/VAE hybrids (Table 1).

## 3 METHODOLOGY

Consider a forward model defined as:

$$d = \mathcal{F}(m, e) + \eta,$$

where $m$ represents model parameters sampled from their prior distribution, $e$ denotes experimental conditions or design parameters, and $\eta$ is random noise sampled from a predefined noise distribution.

The Bayesian inverse problem aims to infer unknown/unobservable parameters $m$ using known experiment parameters $e$ and observations $d$ from the forward model. The solution is characterized by a posterior probability distribution, with density given by Bayes' law:

$$\pi(m|d, e) = \frac{\pi(d|m, e)\pi(m)}{\pi(d|e)},$$

where $\pi(m)$ is the prior distribution encoding prior knowledge about parameters, $\pi(d|m, e)$ is the likelihood, and $\pi(m|d, e)$ is the posterior distribution.

The primary objective is to solve the inverse problem: given observations $d$ and experiment parameters $e$, infer the model parameters $m$. Since $m$ is not uniquely determined by $d$ and $e$, it is characterized by the conditional distribution $\pi(m|d, e)$. The solution can be reformulated as learning the **conditional distribution** $\pi(m|d, e)$.

To achieve this, we employ the conditional flow matching (CFM) framework from [Lipman et al., 2023] (Algorithm 1). This involves first sampling from an unconditional prior distribution for $m$ (denoted as $m_0$).

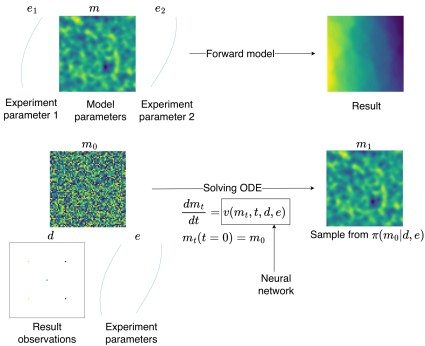

Figure 1: Solving the inverse problem using flow-matching scheme

**Training** We define a conditional interpolation path between $(m_0, d, e)$ and $(m, d, e)$, where $(d, m, e)$ is drawn from the dataset (Appendix B).

In the CFM approach, we learn a velocity field $v_\theta(m_t, t, d, e)$ that minimizes:

where the interpolation path is given by:

$$m_t = (1 - t)m_0 + t \cdot m, \quad t \in [0, 1]$$

Here, $v_\theta$ is a learnable function parameterized by $\theta$ that predicts the velocity field given inputs $(m_t, t, d, e)$. The input dimensions correspond to $m_t$, $t$, $d$, and $e$, while the output dimension matches that of $m$.

During training, elements $(d, m, e)$ are sampled from the dataset, and $m_0$ is drawn from the prior for each iteration of a stochastic optimizer. A neural network effectively represents $v_\theta$ in our experiments.

Once trained, samples from $\pi(m|d, e)$ are generated by solving an ordinary differential equation (ODE) parameterized by the learned velocity field.

A key feature of our approach is the ability to handle arbitrary numbers of observations $d$ and design parameters $e$ as input. This capability stems from our transformer architecture, shown in Figure 5.

## 4 NUMERICAL EXPERIMENTS

We utilize numerical experiment formulations adopted from [Koval et al., 2024]. Specifically, we consider solutions of ordinary differential equation systems modeling disease propagation, as well as elliptic partial differential equations

| Method | Base model | Exact likelihood estimation | No middle-man Training | Arbitrary number of observations |
|---|---|---|---|---|
| MDGM | VAE based on CNN | ✗ | ✓ | ✗ |
| MCGAN | MCMC + GAN | ✗ | ✗ | ✗ |
| PI-INN | PI + flow-based model | ✓ | ✓ | ✗ |
| CFM-Tr (ours) | CFM + Transformer | ✓ | ✓ | ✓ |

Table 1: Comparison of methods for solving Bayesian Inverse problems. *MDGM use the PDE solution as a holistic observation; the problem was not formulated as the recovery of the forward model from a small number of observations

$$\mathbb{E}_{t\sim\mathcal{U}(0,1)}\mathbb{E}_{m_0\sim\text{prior}}\mathbb{E}_{(m,d,e)\sim data}\left[\left\|v_\theta(m_t,t,d,e)-(m-m_0)\right\|^2\right]\to\min_\theta$$

such as the Darcy Flow. These problem classes are widely employed in the literature on Bayesian inverse problems.

## 4.1 SIMPLE NONLINEAR MODEL

After 10,000 runs of the trained model, the generation error is $1.5\cdot10^{-3}\pm0.9\cdot10^{-3}$. Figure 4 in Appendix B shows example paths as we move from the prior distribution to the target distribution $\pi(d,m,e)$. Notably, due to the efficient learning of Flow Matching, the paths are almost straight, indicating optimal transport.

## 4.2 SEIR DISEASE MODEL

The SEIR (Susceptible-Exposed-Infected-Removed) model is a mathematical framework used to simulate the spread of infectious diseases. In this case study, we simulate a realistic scenario where we measure the number of infected and deceased individuals at random times and use this information to recover the control parameters of the ODE system.

For $\mathbf{m_{true}}=[0.4,0.3,0.3,0.1,0.15,0.6]$, after 1,000 calculations the average error is $2.05\%\pm1.04\%$ using a 4-point multilayer perceptron (MLP) model.

## 4.3 PERMEABILITY FIELD INVERSION

We next consider the problem of solving a two-dimensional elliptic PDE. This type of problem is common in the oil industry, where pressure observations from a small number of wells are used to reconstruct the permeability field of an oil reservoir. The equation also has applications in groundwater modeling and many other domains.

Our results show that we can effectively recover the PDE coefficient using just a few strategically placed measurement points. Figure 3 demonstrates that with 8 relatively uniformly distributed points over the solution field, we can obtain an almost identical solution (approximately 2.75%

relative error). The ensemble-generated $\log\kappa$ represents the mean of 50 parameter predictions from the transformer's inference.

## 5 RESULTS AND DISCUSSIONS

Table 2 presents the results of numerical experiments for our proposed method using the following error metric:

$$\varepsilon=\frac{\|\text{DE}(m)-\text{DE}(\tilde{m})\|}{\|\text{DE}(m)\|}$$

where DE represents the solution of the differential equation (ODE or PDE) using either the true parameters $m$ or the generated parameters $\tilde{m}$, computed as an average over 10 generations from the flow matching model.

The true solution of the ODE system and the reconstructed parameter distribution, obtained using only four observation points, are illustrated in Figure 2.

Table 2: The relative inference error of the trained model for two numerical experiments

| N | SEIR Problem 4.2 | Permeability Field 4.3 |
|---|---|---|
| 2 | $10.88\%\pm2.39\%$ | $28.84\%\pm3.43\%$ |
| 3 | $3.31\%\pm1.47\%$ | $16.23\%\pm1.53\%$ |
| 4 | $2.80\%\pm1.37\%$ | $17.80\%\pm1.99\%$ |
| 5 | $2.15\%\pm0.99\%$ | $16.86\%\pm1.76\%$ |
| 6 | $1.97\%\pm0.91\%$ | $7.21\%\pm1.26\%$ |
| 7 | $1.59\%\pm0.75\%$ | $7.48\%\pm1.23\%$ |
| 8 | $1.48\%\pm0.71\%$ | $2.75\%\pm0.60\%$ |

We compare our method against the Metropolis-Hastings MCMC (MH-MCMC) algorithm, running it with sufficient iterations to match the error levels shown in Table 4. The results for the SEIR problem are presented in Table 2.

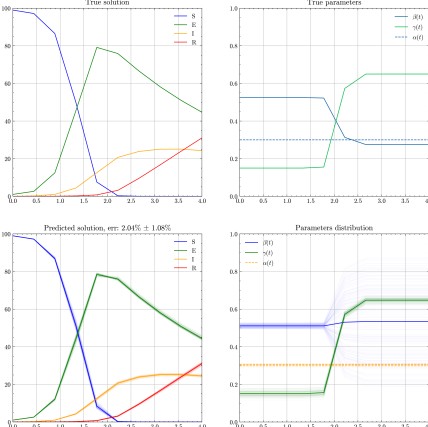

Figure 2: Probabilistic solutions to the inverse problem for $\mathbf{m_{true}} = [0.4, 0.3, 0.3, 0.1, 0.15, 0.6]$

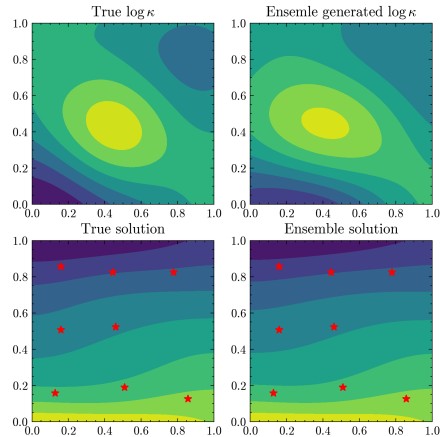

Figure 3: PDE coefficient and solution: true (left) and reconstructed using Flow Matching (right)

**Comparison with MCMC** Conditional Flow Matching (CFM) dominates a tuned Metropolis–Hastings (MH) baseline in both accuracy and speed. On the permeability–inversion benchmark, MH requires $10\,000$ iterations ($\approx 37\text{min}$) yet still exceeds $30\%$ relative $\mathrm{L}_2$ error for $\geq 6$ observations, whereas a single CFM forward pass ($1.08$ s, CPU) yields $2\text{–}8\%$ error (Tables 2–4). For the SEIR model, the CFM Transformer answers in $0.22$ s with analogous gains. More details on the results of the comparison with MCMC can be found in the table below 4

1. **Variable-length conditioning.** Rotary-augmented Transformers generalise to sequences longer than those seen in training, sustaining accuracy as observations accumulate.
2. **Structured transport.** Learned trajectories from prior to posterior are nearly linear (Fig. 4), indicating efficient exploration of parameter space.
3. **Data–efficiency.** Error decreases monotonically with additional observations (Table 2), confirming robustness in dense-data regimes.

Additionally, Conditional Flow Matching shows promise for

determining optimal experiment design parameters $e$, which could further enhance its applicability to practical scientific applications.

# 6 LIMITATIONS AND FUTURE WORK

CFM is only one member of a broader class of generative methods; normalising flows, tensor decompositions Koval et al. [2024], and GANs offer complementary inductive biases that merit systematic comparison. Two challenges stand out. Scaling CFM to very large parameter spaces will likely require architectural changes and significantly larger training corpora. Bayesian Optimal Experimental Design (BOED) hinges on fast, accurate log-likelihood estimates. Although feasible in principle, we have not yet characterised the computational cost or numerical stability of evaluating $\log p(d \,|\, m)$ under a CFM model.

Beyond these issues, tighter coupling to the forward model could improve reliability: after an initial CFM draw $\hat{m}$, a lightweight correction step could enforce consistency with the observations. Finally, the stochasticity of the learned velocity field $v_t$ deserves scrutiny. When the data fully identify the parameters, $v_t$ may collapse to a deterministic mapping, turning CFM into a regression engine. Future work will (i) quantify when noise remains informative, and (ii) compare two practical strategies—sampling diverse noise realisations versus selecting samples that minimise data misfit—to reconcile the discrepancies we observed between CFM and MCMC posteriors.

# 7 CODE AND AVAILABILITY

Technical training details (architectures, learning rates, etc.) are given in Appendix B. The code is written using `PyTorch` framework and is publicly available at

```
https://github.com/
FlowMatchingInverseProblems/
Bayesian-Inverse-Meet-FM
```

# 8 CONCLUSIONS

We believe that our method is quite universal and can be adapted to a large number of problems in a short time when the problem is reduced to a standard Bayesian inverse problem formulation, since it can learn complex nonlinear distributions. Another advantage is the possibility of using an input that is not fixed in terms of the number of observations, where increasing the number of observed points improves accuracy in recovering the solution from the generated parameters. Finally, we can use the learned distribution to do Bayesian optimal experiment design.

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

# Bayesian Inverse Problems Meet Flow Matching: Efficient and Flexible Inference via Transformers
# (Supplementary Material)

## A NUMERICAL EXPERIMENTS PROBLEM STATEMENTS

### A.1 SIMPLE NONLINEAR MODEL

In our experiments, we used the following forward model from Koval et al. [2024]:

$$d(e, m) = e^2 m^3 + m \exp\left(-|0.2 - e|\right) + \eta$$

where $\eta$ follows a known noise distribution, specifically $\mathcal{N}(0, \sigma^2)$. In the simplest example from Koval et al. [2024], the model parameter $m$ is one-dimensional, uniformly distributed on $[0, 1]$. The experiment parameter $e$ is also one-dimensional from $[0, 1]$ and uniformly distributed. We generate random triples $(d_i, m_i, e_i)$ by:

- Sampling $m$ from $\mathcal{U}[0, 1]$
- Sampling $e$ from $\mathcal{U}[0, 1]$
- Sampling noise $\eta$ from the noise distribution
- Computing $d = f(m, e) + \eta$

After sampling, we obtain a dataset in the form of an $N \times k$ matrix, where $k = 3$. These are samples from the **joint distribution** $\pi(d, m, e)$. The prior distribution for training conditional flow matching was a simple uniform distribution $m_0 \sim \mathcal{U}[0, 1]$.

### A.2 SEIR DISEASE MODEL

Following [Koval et al., 2024], we adopt the SEIR model, which assumes a constant population size and is described by the following system of ordinary differential equations:

$$\frac{dS}{dt} = -\beta(t)SI, \frac{dE}{dt} = \beta(t)SI - \alpha E$$
$$\frac{dI}{dt} = \alpha E - \gamma(t)I, \frac{dR}{dt} = \gamma(t)I$$

where $S(t)$, $E(t)$, $I(t)$, $R(t)$ denote the fractions of susceptible, exposed, infected, and removed individuals at time $t$, respectively. These are initialized with $S(0) = 99$, $E(0) = 1$, $I(0) = R(0) = 0$.

The parameters to be estimated are $\beta(t)$, $\alpha$, $\gamma^r$, and $\gamma^d(t)$, where the constants $\alpha$ and $\gamma^r$ denote the rates of susceptibility to exposure and infection to recovery, respectively. To simulate the effect of policy changes or other time-dependent factors

*Submitted to the 41$^{st}$ Conference on Uncertainty in Artificial Intelligence* (UAI 2025). **To be used for reviewing only**.

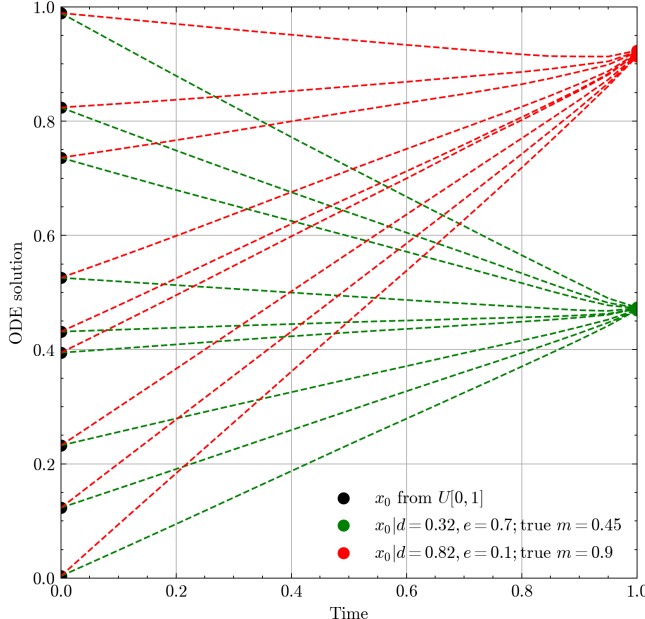

Figure 4: Generation paths of variable $m$ conditional on different $d$, $e$ from prior uniform distribution

(e.g., quarantine and hospital capacity), the rates at which exposed individuals become infected and infected individuals perish are assumed to be time-dependent and parametrized as:

$$\beta(t) = \beta_1 + \frac{\tanh(7(t-\tau))}{2}(\beta_2 - \beta_1)$$
$$\gamma(t) = \gamma^r + \gamma^d(t)$$
$$\gamma^d(t) = \gamma_1^d + \frac{\tanh(7(t-\tau))}{2}(\gamma_2^d - \gamma_1^d)$$

where the rates transition smoothly from initial rates ($\beta_1$ and $\gamma_1^d$) to final rates ($\beta_2$ and $\gamma_2^d$) around time $\tau > 0$.

We fix $\tau = 2.1$ over a time interval of $[0, 4]$. The experiment consists of choosing four time points $e = [a_1, a_2, a_3, a_4] \sim \mathcal{U}[1, 3]$ to measure the number of infected and deceased individuals $d_i = [I_{e_i}, R_{e_i}]$ for $i \in [1, 4]$ ($d \in \mathbf{R}^{2 \times 4}$). The goal is to optimally infer the 6 rates $\mathbf{m} = [\beta_1, \alpha, \gamma^r, \gamma_1^d, \beta_2, \gamma_2^d]$. After training an MLP and solving the flow matching problem, we learn a smooth transition from the distribution $\mathcal{U}[0, 1]^6$ to the distribution $\hat{\mathbf{m}} \sim \rho(\mathbf{m}|\mathbf{e}, \mathbf{d})$.

To summarize the inputs and outputs:

- $e = [a_1, a_2, a_3, a_4] \sim \mathcal{U}[1, 3]$: random measurement times

- $d_i = [I_{e_i}, R_{e_i}]$ for $i \in [1, 4]$ ($d \in \mathbf{R}^{2 \times 4}$): numbers of infected and deceased individuals

- $m = [\beta_1, \alpha, \gamma^r, \gamma_1^d, \beta_2, \gamma_2^d]$: ODE model parameters

Using $\hat{m}$, we can obtain the predicted dynamics of infected and deceased individuals $\hat{d}$. We measure accuracy using:

$$\varepsilon = \frac{\|d - \hat{d}\|_2}{\|d\|_2}$$

## A.3 PERMEABILITY FIELD INVERSION

$$-\nabla \cdot (\kappa \nabla u) = 0$$

with boundary conditions:

$$u(x = 0, y) = f(y, e_1) = \exp\left(-\frac{1}{2\sigma_w}(y - e_1)^2\right)$$

$$u(x = 1, y) = g(y, e_2) = -\exp\left(-\frac{1}{2\sigma_w}(y - e_2)^2\right)$$

The equation is solved using the finite element (FE) method with second-order Lagrange elements on a mesh of size $h = \frac{1}{64}$ in each coordinate direction, where $\kappa$ is a custom 2D matrix. The discretization follows Dolgov et al. [2012].

In this example, the inverse problem consists of estimating the spatially-dependent diffusivity field $\kappa$ given pressure measurements $u$ at pre-determined locations $(x_i, y_i) \in \Omega$. To ensure $\kappa$ is nonnegative, we impose a Gaussian prior on the log diffusivity, $m = \log(\kappa) \sim N(0, C_{pr})$, with covariance operator $C_{pr}$ defined using a squared-exponential kernel:

$$c(x, z) = \sigma_v^2 \exp\left[\frac{-\|x - z\|^2}{2\ell^2}\right] \quad \text{for } x, z \in \Omega,$$

with $\sigma_v = 1$ and $\ell^2 = 0.1$. Using a truncated Karhunen-Loève expansion of the unknown diffusivity field yields the approximation:

$$m(x, \mathbf{m}) \approx \sum_{i=1}^{n_m} m_i \sqrt{\lambda_i} \phi_i(x),$$

where $\lambda_i$ and $\phi_i(x)$ denote the $i$-th largest eigenvalue and eigenfunction of $C_{pr}$, respectively, and the unknown coefficients $m_i \sim \mathcal{N}(0, 1)$. The Karhunen-Loève expansion is truncated after $n_m = 16$ modes, capturing 99 percent of the weight of $C_{pr}$.

The transformer architecture accommodates various input formats for this inverse problem. Here, in addition to the observed solution values, we use the coordinates of measurement points. The specific architecture is detailed in Figure 6.

The input consists of a vector of values $d$ of arbitrary length and two corresponding vectors of coordinates $x, y$. The final input is a matrix $\mathbf{D} = (\mathbf{d}, \mathbf{x}, \mathbf{y})^T$ with shape $(n, 3)$.

## B  TECHNICAL DETAILS

**Dataset**  The key idea is that we can easily sample from the joint distribution $(m_i, d_i, e_i)$. In order to do that, we generate random model parameters (from the prior distribution) and random observation points. When a pair $m_i, e_i$ is given, we can compute $d_i$ using the forward model. Importantly, $m_i$ is also a sample from the conditional distribution $\pi(m|d_i, e_i)$. Thus, when a forward model and prior distributions of model parameters $m$ and experimental parameters $e$ are known, we generate training data by sampling multiple variants of $m$ and $e$ and computing the forward model to obtain observations $d$. For each model parameter $m_i$ we sample $d_i$ for several points $e_i$, thus, our training data consists of tuples of the form $(m_i, d_i, e_i)$, where $d_i$ and $e_i$ may have variable lengths. The model should be able to sample $m_i$ given observations $(d_i, e_i)$. In order to do that, we utilize CFM.

**Architecture**  We parameterize the velocity field using a transformer architecture with bi-directional attention, motivated by the Diffusion Transformer Peebles and Xie [2022]. Specifically, our transformer implementation uses linear projection of input parameters into the embedding space. Time is encoded using a Timestep Embedder as proposed in [Peebles and Xie, 2022], which ensures proper time representation in the embedding space. Root Mean Square (RMS) normalization stabilizes learning dynamics. The activation function is $x = \text{ReLU}(x)^2$. Self-attention uses rotary position embeddings (RoPE), enabling the transformer to learn relative token positions and generalize to sequences longer than those seen during training.

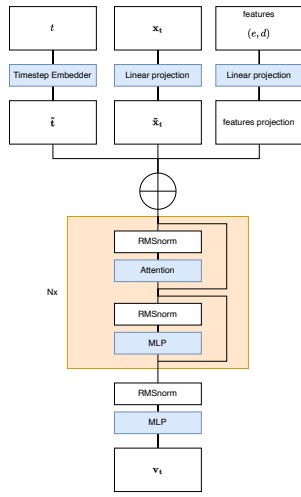

The architecture varies slightly across tasks to accommodate different input data representations. Specific implementations for tasks from Section 4 are detailed in Figure 6 in Appendix B.

Model inference follows Algorithm 2, where the trained CFM model serves as the velocity field in the ODE.

The transformer architecture for two numerical experiments

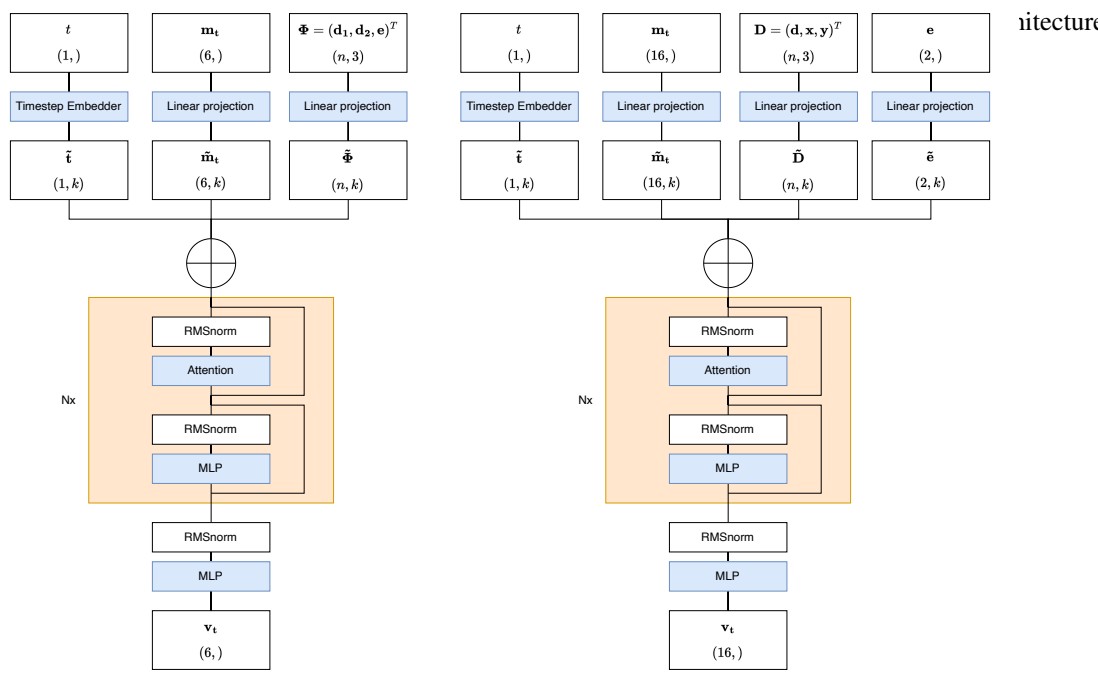

Figure 6: Transformer architecture for 4.2 (left) and 4.3 (right)

Table 3: Hyperparameters for SEIR and Permeability Inversion tasks

| Parameter | SEIR | Permeability Inversion |
|---|---|---|
| learning_rate | 8e-4 | 3e-4 |
| n_emb | 32 | 32 |
| n_head | 4 | 4 |
| n_layer | 6 | 4 |

# C  ALGORITHMS

## C.1  CONDITIONAL FLOW MATCHING

This section provides pseudocode for the core training and inference procedures used in our Conditional Flow Matching (CFM) framework. These algorithms form the backbone of our method for solving inverse problems in various scientific settings.

Algorithm 1 details the training procedure for the conditional flow model. Given a dataset of paired samples and conditioning information, the model is trained to approximate the velocity field that defines an interpolation between prior and posterior samples. The training objective minimizes the squared error between the predicted velocity and the ground-truth velocity vector defined by the linear interpolation between samples.

---

**Algorithm 1:** Conditional Flow Matching Training Algorithm

---

**Input:** Dataset of paired samples $(m_1, e, d)$, neural network model $\mathbf{v}_\theta(t, m, e, d)$, conditioning data $e$ and $d$, time $t \sim \text{Uniform}(0, 1)$, number of epochs $N_{\text{epoch}}$

**Output:** Trained conditional flow model $\mathbf{v}_\theta(t, m, e, d)$

**for** 1 *to* $N_{epoch}$ **do**

    **for** *each minibatch of samples $(m_0, m_1)$* **do**

        $t \sim \mathcal{U}(0, 1)$                                              `// Sample t`

        $m_0 \sim$ prior distribution

        $m_t \leftarrow t \cdot m_1 + (1 - t) \cdot m_0$

        Compute the target velocity: $u_t \leftarrow m_1 - m_0$

        Predict the velocity: $v_t \leftarrow \mathbf{v}(t, m_t, e, d)$

        Compute the loss: $\mathcal{L}(\theta) \leftarrow \mathbb{E}\left[\|v_t - u_t\|^2\right]$

        Compute gradients: $\nabla_\theta \mathcal{L}(\theta)$

        Update $\theta$ using the optimizer and $\nabla_\theta \mathcal{L}(\theta)$

    **end**

**end**

**return** $\mathbf{v}_\theta(t, x, e, d)$

---

Algorithm 2 presents the inference procedure. After training, the model is used to define a deterministic flow by solving an ordinary differential equation (ODE) starting from a sample from the prior distribution. The terminal state of this ODE corresponds to a sample from the conditional distribution given the observations and experimental conditions.

Together, these two procedures enable the model to learn and sample from complex conditional distributions without relying on stochastic sampling or iterative optimization during inference.

---

**Algorithm 2:** Conditional Flow Matching Inference Algorithm

---

**Input:** Trained CFM model $\mathbf{v}_\theta(t, x)$, conditioning data $e$ and $d$, initial sample $x_0$, experiment parameters $e$, arbitrary observations $d$

**Output:** Generated parameters $m$

$x(t = 0) \sim$ prior distribution

$x(t = 1) \leftarrow \text{Solution} \frac{dx}{dt} = \mathbf{v}_\theta(t, x_t, e, d)$

**return** $x(t = 1)$

---

## C.2 RELATIVE ERRORS FOR NUMERICAL EXPERIMENTS USING MCMC

Table 4: Relative errors for numerical experiments using MCMC

| N | SEIR Problem | | Permeability field | |
|---|---|---|---|---|
| | $N_{sample}$ | Relative Error | $N_{sample}$ | Relative Error |
| 2 | 15 000 | 31.39% | 10 000 | 56.57% |
| 3 | 10 000 | 3.26% | 10 000 | 39.40% |
| 4 | 5 000 | 3.24% | 10 000 | 57.66% |
| 5 | 5 000 | 2.74% | 10 000 | 55.71% |
| 6 | 5 000 | 1.64% | 10 000 | 42.63% |
| 7 | 5 000 | 4.07% | 10 000 | 36.13% |
| 8 | 10 000 | 1.44% | 5 000 | 60.02% |

