# OpenReview forum: "Bayesian Inverse Problems Meet Flow Matching: Efficient and Flexible Inference via Transformers"
_auai.org/UAI/2025/Workshop/TPM — TPM 2025_

### Official Review · Reviewer_S7N3 · 2025-06-12
**Flow matching for Bayesian inverse problems**

**Rating:** 3

**Review:**

This paper focuses on Bayesian inverse problems. The authors train a conditional flow matching model with Transformer architecture. The flow model enables fast sampling and exact likelihood computation, which show its advantages over other generative models. Experiments are conducted on ODE, PDE systems, where the target is to learn the parameter posteriors. The proposed method outperforms MCMC.

The flow matching models are popular recently. Using it to solve inverse problems is of great interest to the workshop. While this paper presents some preliminary results, I support them to be shared in the workshop.

---

### Official Review · Reviewer_spxv · 2025-06-15
**TPM may not be the best fit**

**Rating:** 1

**Review:**

The paper considers the Bayesian estimation problem of learning the parameters of a model given data, with a prior on the parameters.  They propose to model the posterior distribution of the parameters given the data based on a Conditional Flow Matching approach, which allows for efficient sampling.

There are some numerical experiments showing that the parameters of some simple problems can be recovered, using synthetic data.

One issue I have is that half of the references are arxiv references, and not proper citations.  For example, Kingma & Welling has a proper citation.  For AI papers, an arxiv citation is often viewed as a citation to a pre-print of unknown quality.  (I realize the appropriateness of arxiv citations varies from field-to-field).

I am not sure why the paper claims that formulating the Bayesian inverse problem as learning a conditional probability distribution.  It looks like to me a standard Bayesian inference problem, of identifying the posterior over the parameters given the data.

I would say this paper is somewhat in the scope of TPM, if you view Conditional Flow Matching as sharing comparable goals (I am not familiar with CFM tho).